# Endoprosthetic Reconstruction of the Proximal Humerus with an Inverse Tumor Prosthesis

**DOI:** 10.3390/cancers15225330

**Published:** 2023-11-08

**Authors:** Anna Maria Rachbauer, Kristian Nikolaus Schneider, Georg Gosheger, Niklas Deventer

**Affiliations:** Department of General Orthopedics and Tumor Orthopedics, Muenster University Hospital, 48149 Muenster, Germanyniklas.deventer@ukmuenster.de (N.D.)

**Keywords:** bone tumors, metastases, proximal humerus, inverse tumor prosthesis

## Abstract

**Simple Summary:**

Reconstructing the proximal humerus after tumor resection remains a significant challenge, often resulting in reduced function and a diminished quality of life for patients due to the loss of soft tissues and bone. This research highlights the advantages of using an inverse tumor prosthesis for patients with proximal humerus bone tumors. It offers long-term results that have not been available in the existing literature, addressing concerns related to potential complications such as prosthetic loosening, infection, dislocation, and the loss of function. This study is valuable as it provides insight into an effective reconstruction method for a complex problem, fostering further research refinement in this field.

**Abstract:**

Reconstructing the proximal humerus after tumor removal is challenging due to muscle and bone loss. The current methods often result in poor shoulder function. This study assessed the long-term functional and oncological outcomes of using an inverse proximal humerus prosthesis in 46 patients with bone tumors. The results showed a mean range of motion of 62° in anteversion, 28° in retroversion, and 55° in abduction. Notably, 23 patients achieved over 90° of shoulder abduction, with an average of 140°. The median Musculoskeletal Tumor Society Score was 25. Complications included infection in two radiotherapy patients and single dislocations in seven patients. One patient with recurrent dislocations needed revision surgery. In conclusion, the use of the inverse proximal humerus prosthesis in bone tumor treatment yields excellent shoulder function and high patient satisfaction. This approach is especially beneficial for those with metastatic disease.

## 1. Introduction

The proximal humerus ranks as the third most prevalent site for both primary bone tumors and secondary malignancies, collectively accounting for 7–10% of such cases [1,2]. The treatment approach for primary bone tumors typically involves the wide resection of the tumor, encompassing not only the affected bone but also the surrounding soft tissues, followed by the imperative task of reconstructing the resultant osseous and soft tissue defects. Presently, the gold standard for such reconstruction employs anatomically tailored modular megaprostheses [3,4].

Notably, this approach has gained prominence not only for primary bone tumors but also in the management of patients afflicted with solitary bone metastases, owing to its potential to enhance oncological outcomes [5,6]. However, a significant challenge persists: the extensive loss of soft tissue often compromises shoulder function, even when utilizing attachment tubes in conjunction with anatomically designed prostheses [5,7,8].

Conversely, reverse shoulder arthroplasty has demonstrated remarkable success in patients afflicted by arthropathy stemming from massive rotator cuff tears. This led to the intriguing hypothesis that the fundamental design principles underpinning reverse shoulder prostheses could be applicable in cases involving the resection of the proximal humerus. At the Department for Orthopaedic Surgery of the University of Münster, an innovative inverse tumor prosthesis was conceived and successfully introduced [4,6,9].

Despite the promising initial results reported from a small patient cohort, there remains a noticeable dearth of data from a bigger group to underline these first findings, as well as of long-term data from a more expansive patient population. Therefore, this study aims to fill this knowledge gap by presenting functional outcomes from a substantially larger patient group, surpassing what has been documented in the existing literature. Additionally, the study endeavors to provide comprehensive long-term results (with follow-up periods exceeding 24 months) for a sub-cohort, addressing various aspects including shoulder function, prosthetic biomechanics, residual pain, emotional well-being, as well as the impact of the primary disease and associated therapies.

This collective insight promises to elucidate the true potential and suitability of the inverse proximal humerus prosthesis in the complex landscape of tumor-related humeral defects.

## 2. Materials and Methods

Between the years 2003 and 2021, a total of 60 patients underwent proximal humerus resection and subsequent replacement with the inverse proximal humerus tumor endoprosthesis (IPHP)—specifically, the MUTARS Inverse, manufactured by Implantcast GmbH, Buxtehude, Germany—at our institution. This is a retrospective study with a meticulous review of the medical records conducted for each patient, encompassing the demographics, tumor characteristics, histological findings, details of surgical and oncological interventions, complications and their respective management, occurrences of local recurrences, and radiographic evaluations.

Functional assessments and the Musculoskeletal Tumor Society (MSTS) score were performed at the latest follow-up. The MSTS score serves as a reliable tool for assessing and comparing the functional and emotional outcomes of patients post-tumor resection and musculoskeletal reconstruction. This comprehensive score evaluates parameters such as pain, functional capacity, emotional acceptance, hand positioning, manual dexterity, and lifting ability, assigning numerical values ranging from 0 to 5. The final score is categorized as excellent (≥23 points), good (15–22 points), fair (8–15 points), or poor (<8 points).

Out of the initial cohort of 60 patients, 14 individuals were excluded from the functional analysis. This exclusion was prompted by various factors, including insufficient follow-up duration in five cases, six patients succumbing to the disease within three months following surgical intervention, one patient undergoing ongoing revision due to infection, and two patients excluded from functional analysis due to the presence of a permanent spacer following periprosthetic infection (Figure 1).

To enhance joint stability and facilitate active function through the reattachment of soft tissues to the implant—notably the deltoid, latissimus dorsi, subscapularis, and pectoralis muscles—an attachment tube was employed for all patients during surgery. Subsequently, the shoulder was immobilized in a sling with an abduction pillow for a duration of four to six weeks, contingent upon the extent of soft tissue reconstruction (Figure 2).

The initial follow-up, comprising clinical evaluations and X-ray examinations, occurred three months post-surgery, with subsequent follow-up intervals tailored to the nature of the primary disease. For malignant bone tumors, three-month intervals were maintained for two years, transitioning to six-month intervals until the fifth year, and then annually. Conversely, patients with metastatic diseases underwent yearly assessments after the initial three-month control. Multimodal treatment, including chemotherapy and/or radiotherapy, was administered in accordance with the recommendations of the local tumor board. Notably, patients below the age of 65 diagnosed with osteosarcoma and high-grade spindle cell sarcoma received neoadjuvant and adjuvant chemotherapy, adhering to the prevalent protocols established by the Cooperative Osteosarcoma Study Group (COSS-96, EURAMOS-1, EURO-B.O.S.S.). Similarly, patients with Ewing sarcoma were managed in line with protocols outlined by the Cooperative Ewing Sarcoma Study Group (EURO-E.W.I.N.G.99, Ewing 2008). Radiation therapy was considered for Ewing sarcoma patients displaying inadequate responses to chemotherapy and for patients with spindle cell sarcomas or metastases subsequent to pathological fractures.

Based on histopathological records, wide resection was successfully achieved for all primary malignant bone tumors. Marginal resection was consistently performed for metastatic cases, although it is worth noting that metastases presenting with pathological fractures were categorized as intralesional resections, and radiotherapy was prescribed as a mandatory adjunct.

Statistical analysis was conducted using Microsoft Excel for Mac (Version 15.31) and SPSS (Version 27, IBM, Endicott, NY, USA). The Kolmogorov–Smirnov test was utilized to assess data distribution, with nonparametric analyses executed via the Mann–Whitney U-test, and parametric comparisons made using the *t*-test. All *p*-values were calculated as two-sided, with a threshold of *p* < 0.05 considered statistically significant.

Ethical approval for this study was diligently secured from the local ethics committee (Ethik-Kommission Westfalen-Lippe, reference number: 2018-199-f-S).

## 3. Results

### 3.1. Demographics and Tumor Characteristics (Table 1)

The median age of the 46 patients included in this study at the time of surgery was 52 years, ranging from 12 to 77 years. The median follow-up duration was 25 months, with a range spanning from 3 to 176 months. Among the patients, 25 underwent wide resection due to primary malignant bone tumors, while 21 patients with bone metastases received marginal resection. In cases involving pathologic fractures, intralesional resection was performed. The axillary nerve was successfully preserved in 43 patients, with partial resection observed in four cases. Postoperative radiation therapy was administered to 20 patients. Notably, two of these patients received radiotherapy preoperatively: one for multiple myeloma and the other for a hemangiopericytoma.

**Table 1 cancers-15-05330-t001:** Patient demographics, oncologic, and surgical details.

Variable	*n* (%)
included patients	46
female	15 (32%)
male	31 (68%)
median follow-up	25 months (IQR 8–176)
deaths during observation	13 (28%)
age	52 years (IQR 12–81)
tumor histology	
primary bone tumor	25 (54%)
osteosarcoma	8 (17%)
chondrosarcoma	9 (19.5%)
Ewing’s sarcoma	7 (15%)
giant cell tumor	1 (2.5%)
metastases	21 (46%)
pathologic fracture	8 (17%)
adjuvant treatment	27 (58%)
radiotherapy	20 (43%)
preoperative	3 (6%)
postoperative	17 (37%)
chemotherapy	27 (58%)
preoperative	15 (32%)
postoperative	12 (26%)

In terms of tumor involvement, 30 patients exhibited tumors affecting both bone and soft tissues, while 16 patients had tumors limited to the bone. The median length of resection was 14 cm, ranging from 7 to 34 cm. In three cases, total humerus replacement was necessitated.

### 3.2. Complications and Revision Surgery

Two patients who experienced prosthetic dislocation chose not to undergo revision surgery due to their pain-free status and acceptance of limited function. One patient with recurrent dislocations underwent revision surgery, ultimately requiring a Bateman procedure. While the range of motion was limited, it was deemed satisfactory by the patient.

An additional three patients experienced single dislocations, which were successfully managed through repositioning and subsequent immobilization in a shoulder brace for four weeks, followed by intensive physiotherapy. All three patients achieved a stable situation with good function.

Two patients suffered from recurrent dislocations, with one undergoing surgical revision, resulting in the prevention of further dislocation events. The other patient, after the rupture of the attachment tube, experienced recurrent dislocations but opted against surgical intervention due to good function and painless, spontaneous repositioning.

As described above in two cases, the prosthesis had to be removed and replaced with a spacer due to infection. In both instances, the patients declined further replacement with a proper prosthesis, and the situation was considered stable. Those patients, as well as the one who is undergoing ongoing infection therapy were excluded from the statistical analysis of the functional outcome. Nonetheless, those three patients still showed a good MSTS with an average of 26.

Overall, complications, including dislocation or infection, were encountered in 20% of the patients but were satisfactorily managed.

### 3.3. Radiographic Evaluation

Throughout the follow-up period, radiographic assessments consistently revealed stable prosthetic fixation, with no evidence of implant loosening, either in the stem or the metaglene (glenoid component).

### 3.4. Range of Motion and MSTS Score

The range of motion data demonstrated the following findings: anteversion measured at 62° (range: 10°–180°), retroversion at 28° (range: 0°–170°), and abduction at 55° (range: 5°–90°). Remarkably, 23 patients (approximately 50%) achieved active shoulder abduction exceeding 90°, with a median abduction angle of 140° (range: 95°–180°).

The median Musculoskeletal Tumor Society (MSTS) score was 25, with a range of 14 to 30.

### 3.5. Effect of Radiotherapy

Significant differences were observed in the range of motion between patients who underwent radiotherapy and those who did not. Specifically, anteversion was significantly reduced in patients who received radiotherapy (mean 50°) compared to those without radiotherapy (mean 70°), with a *p*-value of 0.05 (Figure 3). However, abduction did not show significant differences between the two groups (56° versus 50°). Regarding the ability to abduct the arm over 90°, 40% of patients who underwent radiotherapy achieved this compared to 42% of those who did not, with a mean of 12° in the radiotherapy group and 30° in the non-radiotherapy group. Notably, despite differences in function, the MSTS score did not significantly differ between the two groups, with scores of 24 in the radiotherapy group and 25 in the non-radiotherapy group.

### 3.6. Comparison by Tumor Type and Resection Extent

Significant differences were also identified when comparing the patients who underwent surgery for primary malignant bone tumors (25 patients, median age 38 years) with those who required surgery for metastatic disease (21 patients, median age 65 years). The patients with primary malignant tumors demonstrated superior anteversion (72° in primary malignant tumors versus 49° in metastatic disease, *p* = 0.04) and retroversion (31° versus 24°, *p* = 0.0006) (Figure 3). While abduction did not exhibit significant differences (59° in primary malignant tumors versus 49° in metastatic disease, *p* = 0.14), it was notably better in primary malignant tumors regarding the ability to abduct the arm over 90° (28° versus 16°). Specifically, 48% of the primary malignant tumor patients could achieve this compared to 34% of those with metastatic disease. The MSTS score was equally satisfactory, with scores of 24 in both groups.

Moreover, the patients with resection restricted to bone (BR) demonstrated superior function compared to those requiring bone and soft tissue removal (BSR). The median resection length in the BR group was 13.5 cm (ranging from 7 to 24.5 cm), in the BSR group 16 cm (ranging from 10 to 34 cm). Specifically, the BR patients exhibited better anteversion (84° in BR versus 50° in BSR, *p* = 0.01), retroversion (35° versus 25°, *p* = 0.04), and abduction (67° versus 48°, *p* = 0.02) (Figure 3). A notable difference was also observed in the ability to abduct the arm over 90°, with 62.5% of the BR patients achieving this compared to 40% of the BSR patients. The MSTS score was consistently satisfactory, with scores of 26 in the BR patients and 24 in the BSR patients.

### 3.7. Longer-Term Follow-Up

A subset of 26 patients had follow-up durations exceeding 24 months, with a median follow-up duration of 66 months (range: 24–176 months) (Figure 2C and Figure 4). During this extended follow-up period, one patient experienced infection, while four patients encountered dislocations. Importantly, the dislocations of the humeral implant occurred exclusively during the first year post-surgery. Following repositioning and conservative treatment, all patients regained function. In one case, the polyethylene liner had to be replaced nine years after the initial implantation due to perceived instability. Subsequent revision surgery successfully restored the range of motion to previous levels. At the latest follow-up, radiological assessments showed no signs of humeral or glenoid component loosening. Nevertheless, in 62% of the patients, stress shielding was seen, but so far, complications due to the stress shielding has not occurred.

The average MSTS score was 24.5 (82%).

## 4. Discussion

Reconstruction following the resection of malignant tumors in the proximal humerus presents a multifaceted challenge, and various options have been explored. Over recent years, the utilization of endoprosthetic devices has become increasingly prevalent due to their lower complication rates compared to biological reconstruction [5,6,10,11]. Among the available choices, anatomical prostheses have been most commonly employed. However, even when supplemented with reconstructive procedures, like the Bateman technique, functional outcomes have often fallen short of satisfaction [7,12,13]. In light of these limitations, reverse shoulder arthroplasty has gained prominence [4,6].

A notable concern, particularly in younger patients, is the potential loosening of the glenoid component, which can be a significant drawback [14]. In our study, the extended follow-up of more than two years for some patients did not reveal any instances of glenoid component loosening, providing reassurance about the long-term stability of this prosthetic approach.

During our analysis of the X-rays, the sole noteworthy observation was the presence of stress shielding. This phenomenon, akin to the findings of Klingebiel et al. [2], who explored it in the context of modular mega prostheses for the proximal humerus, appeared to be most prominent in the initial year following surgery. Notably, thus far, no complications have arisen as a result of this happening, suggesting that, for the time being, stress shieling may be disregarded. Nevertheless, further research is warranted to substantiate this assertion.

It is well-documented that proximal humerus prostheses are susceptible to higher polyethylene wear in comparison to anatomically shaped implants [15]. Despite this, none of our patients experienced aseptic loosening due to wear-induced inflammatory reactions. Notably, one patient who had been treated for osteosarcoma required an exchange of the polyethylene liner after nine years. This revision was driven by a progressive sensation of instability, possibly attributable to the patient’s high level of sports activity. Subsequent to the revision surgery, the patient was able to resume physically demanding hobbies, including apnea diving, bouldering, and tennis.

To better characterize the reasons behind the potential failures of tumor endoprostheses, Henderson et al. introduced a classification encompassing five types: soft tissue failures (type 1), aseptic loosening (type 2), structural failures (type 3), infection (type 4), and tumor progression (type 5) [16]. In our cohort, the most frequent cause of failure was type 3, involving instability and dislocation, which affected 17% of our patients. This finding aligns with the current literature, where dislocation emerges as the most common complication [6,17]. It is noteworthy that while dislocation rates vary in the literature, reaching up to 45% [6,13,18], our cohort exhibited a lower rate.

Our study revealed that the range of motion in our patients was similar to that reported in other study groups [6,17], and this functional capacity remained unimpaired over time [4,5]. Interestingly, patients whose soft tissues could be preserved demonstrated a significantly better range of motion, an unsurprising finding given the greater number of muscles available for reattachment. A further explanation for the positive functionality lies in the incorporation of a Trevira tube for soft tissue attachment. This method, as demonstrated by Gosheger et al. in 2006 [19], has become a well-established practice in megaprosthesis reconstruction ever since. Moreover, as previously outlined in the literature, a fully functional axillary nerve is imperative for optimal function [4].

Upon the closer examination of the two groups, the improved functionality within the bony resection group can partly be attributed to the shorter resection length (13.5 cm compared to 16 cm). However, a more detailed analysis of individual cases revealed that the patient with the shortest resection (7 cm) in the bony resection group exhibited similar functionality to the patient with the longest resection (34 cm) in the soft tissue group. It is worth noting that the first patient received adjuvant radiotherapy, while the patient with the 34 cm resection only underwent chemotherapy.

Importantly, a lesser range of motion did not appear to significantly impact patient satisfaction, as evidenced by similar MSTS scores in both groups (87% versus 80%).

Similarly, our results showed that the patients who underwent radiation therapy had comparable MSTS scores despite a notable difference in functional outcomes when compared to patients without radiotherapy. This suggests that while patients receiving radiation therapy may experience decreased function, their overall satisfaction remains relatively high. Additionally, the patients suffering from primary bone tumors exhibited better functional outcomes than those treated for bone metastasis. The patients who underwent wide resection for primary malignant bone tumors demonstrated significantly better function, likely attributable to their younger average age (38 years versus 65 years) compared to the metastatic disease cohort. The influence of soft tissue damage due to radiotherapy cannot be discounted as a contributing factor.

In summary, the patients who required the removal of a smaller amount of tissue and did not undergo radiotherapy achieved the best functional outcomes.

Notably, the preservation of the axillary nerve is paramount when considering reconstruction with an inverse tumor prosthesis [4,6,9]. Therefore, the precise planning of the biopsy site is crucial to ensure safe oncological margins [20].

The study’s retrospective design presents a limitation, as does the relatively small patient cohort when compared to studies of other types of endoprostheses, such as modular prostheses for the proximal humerus or megaprotheses of the lower extremity. It is important to note that future research should investigate long-term follow-up results to validate the current findings.

## 5. Conclusions

In conclusion, our study underscores the benefits of utilizing the inverse tumor prosthesis after proximal humerus tumor resection. It delivers good functional results, as long as the axillary nerve can be spared, and exhibits remarkable long-term stability, with no cases of aseptic loosening of the humeral or glenoid component in our extended follow-up patients. This approach is particularly advantageous in metastatic disease, especially in the presence of pathologic fractures. As we continue to collect data and add new cases to our series, the further validation of these findings and the refinement of surgical techniques are essential.

## Figures and Tables

**Figure 1 cancers-15-05330-f001:**
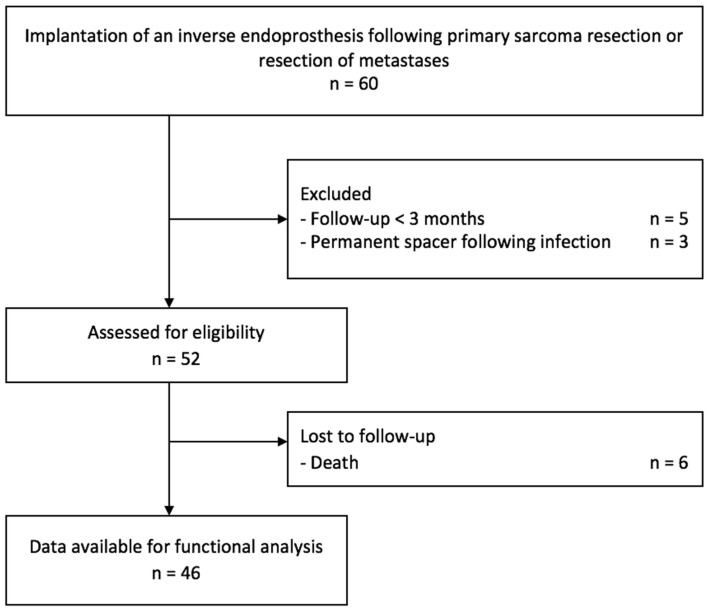
Study flow chart.

**Figure 2 cancers-15-05330-f002:**
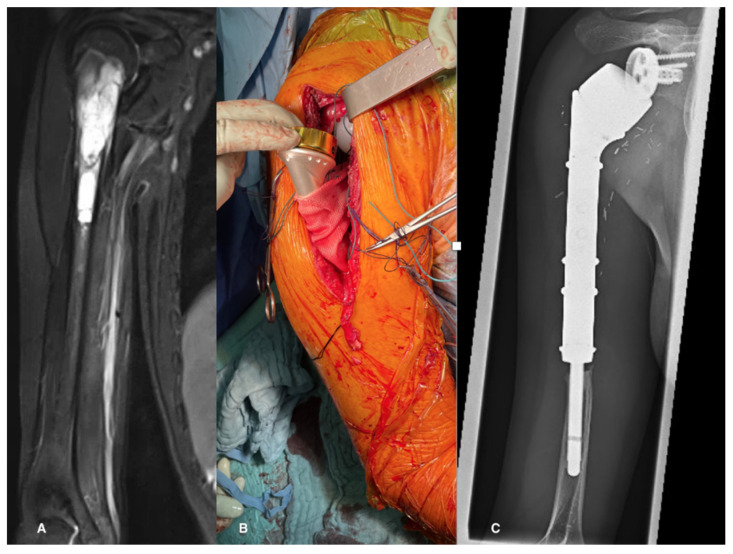
(**A**) MRI showing a Ewing’s sarcoma in a 13-year-old patient, (**B**) intraoperatively after reconstruction with an inverse tumor prosthesis and an attachment tube, (**C**) the 7-year follow-up X-ray shows, besides stress shielding, no signs of implant complications.

**Figure 3 cancers-15-05330-f003:**
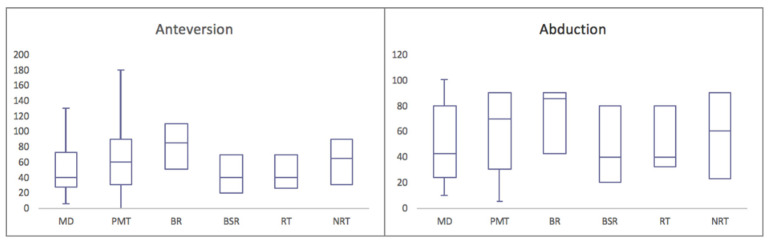
Range of motion in degrees. (MD—metastatic disease, PMT—primary malignant tumor, BR—bony resection, BSR—bone and soft tissue resection, RT—radiotherapy, NRT—no radiotherapy).

**Figure 4 cancers-15-05330-f004:**
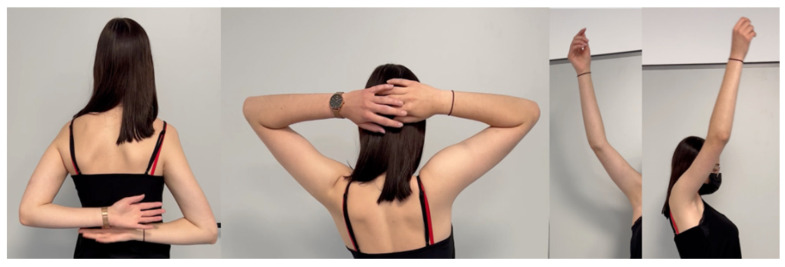
Active range of motion 5 years post-operation of a patient that was treated with Ewing’s sarcoma at the age of 13: neck grip, 180° abduction and anteversion.

## Data Availability

The data presented in this study are available on request from the corresponding author.

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
