# Peer review of "Endoprosthetic Reconstruction of the Proximal Humerus with an Inverse Tumor Prosthesis"

_cancers, 2023, doi:10.3390/cancers15225330_

Round 1

Reviewer 1 Report

Comments and Suggestions for Authors

This is a helpful article with good follow up.

The exclusion criteria do need clarifying.

At lines 73-75 it is stated that 3 patients were excluded from the series due to treatment for infection.

Later in the article, it appears that these 3 patients have in fact been included and this needs clarifying. If they were not included, it would appear that the infection rate was in fact 6 patients out of 60.

The authors point out that the axillary nerve was successfully retained in 43 cases which would suggest that the deltoid muscle was functioning in all of these patients. This may be an explanation of the very good functional results.

The other factor which may account for the good function is the number of patients with purely bone tumours in whom the muscles could be completely preserved and the short resection cases.

Although the authors have looked at function based on a number of factors, have they been able to identify any combinations of factors which led to a particularly good or poor result. For instance, one would expect patients with a pure bony tumour, with a short resection and preservation of the axillary nerve to have almost normal function no matter what implant was inserted whereas long resections with removal of the axillary nerve and soft tissue resection would be expected to have the worst.

Figure 3 gives details of the range of motion but the caption should spell out what the abbreviations are so the reader does not have to refer back to the text.

Figure 2  shows a very successful result 7 years following resection of a Ewing sarcoma which appears to be purely intraosseous. Figure 2C is stated to show no signs of implant complications but in fact, there is marked stress resorption of the bone at the bone / prosthesis interface and also on the lateral cortex. Have the authors observed if this is a common problem and does it get worse with time?

Reference 6 has a typo.

Reviewer 2 Report

Comments and Suggestions for Authors Interesting article on a poorly described topic.

The article concerns endoprosthetic reconstruction of the proximal humerus with an inverse tumor prosthesis.

My comments:

Abbstract

Well and clearly written.

Introduction

Well and clearly written.

A good review of the literature related to the topic.

Please add a clear research hypothesis.

Please change the purpose of the work - You write that you want to evaluate the long-term result, and the average observation period is 25 months. 25 months is not a long-term observation period.

Materials and methods

Please specify whether the study was prospective or retrospective.

Please describe the exact inclusion and exclusion criteria for the study.

Results

Well and clearly written.

Good quality and clear Tables and figures.

Results well presented.

Discussion:

Please describe in detail why You have such good results - such good mobility of the shoulder joint after such an extensive surgery.

 Please outline the limitations of your work.
